# Photovoice Reveals Residents’ Concerns for Air and Water Quality in Industry-Impacted Rural Community

**DOI:** 10.3390/ijerph20095656

**Published:** 2023-04-27

**Authors:** Shelby M. Rimmler, Sarah Shaughnessy, Ellis Tatum, Naeema Muhammad, Shaelyn Hawkins, Alexandra Lightfoot, Sherri White-Williamson, Courtney G. Woods

**Affiliations:** 1School of Social Work, University of North Carolina at Chapel Hill, Chapel Hill, NC 27599, USA; 2Planning & Development, City of Raleigh, Raleigh, NC 27601, USA; 3Independent Researcher, Snow Hill, NC 28580, USA; 4North Carolina Environmental Justice Network, Raleigh, NC 27612, USA; 5Gillings School of Global Public Health, University of North Carolina at Chapel Hill, Chapel Hill, NC 27599, USA; 6Environmental Justice Community Action Network, Clinton, NC 28328, USA

**Keywords:** landfill, water quality, CBPR, environmental justice, rural health

## Abstract

Rural communities of color in the southeastern U.S. experience a high burden of environmental hazards from concentrated industry placement. Community-engaged research and qualitative methods can improve our understanding of meaning-making in a community impacted by polluting facilities. This study applies the photovoice method to assess how a predominantly African American community in rural North Carolina, impacted by a landfill and confined animal feeding operations (CAFOs), perceives their health-related quality of life (HRQoL). Two research questions were developed with community-based partners: (a) How do environmental health concerns in this community influence residents’ perceptions of their HRQoL? and (b) How do community and county factors facilitate or inhibit community organizing around these concerns? Three photo assignment sessions were held to engage participants in discussions related to the research questions. Researchers analyzed discussion audio recordings and identified themes related to concerns about the following issues: health and quality of life, the landfill industry’s influence on community cohesion and self-determination, and actions to address environmental injustice in Sampson County. Photovoice benefits community-engaged researchers by providing a process for assessing the research interests of a community. Photovoice also serves community organizers by providing residents with a structured way to discuss their lived experiences and strategize ways to reduce hazard exposure.

## 1. Introduction

The community-based participatory research (CBPR) and participatory action research (PAR) approaches redefine the conventional roles of researcher and subject by positioning them as co-investigators. CBPR aims to dissolve the researcher–participant power dynamic and utilize all participants’ strengths to effect change [1,2,3]. Qualitative methods, when applied within a CBPR framework, can offer greater context on an issue and shift the perspective by orienting the research towards questions that are most relevant to those impacted by the issue. This is especially useful for environmental health research, which often relies solely on quantitative methods and centers on technical, top-down solutions [4,5,6].

Observations, interviews and focus groups are some of the most commonly used qualitative methods that can be applied to examine community-level impacts of environmental hazards [7]. However, some experiences are hard to express with words alone. Photovoice is a qualitative research method that allows researchers and participants to visualize individuals’ perceptions of their everyday living conditions. Furthermore, photovoice is a CBPR research method that shifts the power into the hands of the community using photographs as inspiration for identifying concerns, facilitating collective discussions, and generating action steps to address the concerns [2,8,9].

By having an opportunity to create and reflect on images representative of the issue, community members can provide unique insights into community challenges and strengths. Photovoice is a useful research process to amplify the voices of traditionally marginalized and disenfranchised populations to promote equity and advocacy for change [10]. As such, it is an appropriate method to employ to explore the perceptions of health-related quality of life (HRQoL) of rural residents and barriers and facilitators to community organizing in a community experiencing environmental injustice. In recent years, photovoice has been used as an effective participatory research tool to address local environmental health issues while achieving other complementary goals such as supporting youth empowerment [11,12,13,14], illuminating indigenous perspectives on environmental change [15,16], highlighting the intersection of environmental, social and political pressures in urban settings [17,18,19], and giving voice to highly marginalized segments of the population, like migrant farmworkers and their families [20,21]. In rural and remote settings, photovoice can be especially important for shedding light on unnoticed and largely misunderstood issues and can enhance civic engagement and participants’ impact on local policy [22,23,24,25].

The area of focus for this photovoice study was Sampson County, NC. Located in the eastern region of the state, Sampson County ranks second in the state for the density of hogs, with over 1.8 million produced annually at 470 facilities across the county [26]. Most facilities house over 2500 hogs (which is the defining size of a confined animal feeding operation, CAFO) and manage the animal waste using an outdated spray and lagoon system where fecal waste is sprayed onto neighboring agricultural land as fertilizer. Previous research has shown that CAFOs contribute to pervasive neighborhood odors and airborne irritants [27,28], more self-reported respiratory symptoms, including asthma symptoms among middle schoolers [29], decreased lung function [30,31], mood disorders [32,33,34,35], and disruptions in engaging in outdoor activities [4,36,37].

In addition to CAFOs, Sampson County also hosts a large regional landfill, which accepts municipal solid waste and construction and demolition waste [38]. Previous research in NC and other states in the southeast has shown that landfill site location affects African American, low-income, and rural communities disproportionately [39,40]. Landfills emit air pollutants that can cause cancer and respiratory disease and may disperse into nearby areas [41,42]. Landfills may also contribute to poor water quality and poor mental health of nearby residents [43,44,45].

The purpose of this study was to assess the HRQoL impact of CAFOs and the Sampson County Regional Landfill on residents of a predominantly African American community in Sampson County, NC. Initial meetings between the researchers and community participants revealed anecdotal descriptions of hazards exposures and general concerns about health impacts from neighboring CAFOs and a landfill and opportunities for capacity building to address those concerns. Community partners had been struggling to maintain momentum and community support for environmental justice organizing and believed the photovoice method could be a beneficial tool to better understand broader community concerns and desires for action. Two research questions were co-developed by the research team and a partnered community organizer from Sampson County to guide project development and data analysis: (a) *How do environmental health concerns in this community impact residents’ perceptions of HRQoL?* and (b) *how do community and county factors facilitate or inhibit constituents’ organization around these concerns?* The photovoice method allowed community members to set the research agenda and bring priority topics related to environmental justice to the forefront of discussion. We believe this method allows for the identification of community concerns and prioritized action responses, which will ultimately lead to more community buy-in and organizing capacity.

Beyond the outcomes of this specific project, photovoice offers a model for community-driven documentation of barriers to healthy living and a basis for grassroots organizing to address community concerns.

## 2. Materials and Methods

### 2.1. Study Location

Sampson County is considered a rural county, with a total population of 59,036 residents, according to the 2020 census [46]. The Sampson County Regional Landfill is located in the Snow Hill community. Participants of the study describe the Snow Hill community as predominantly Black or African American. Compared to the remainder of the county, the census block group encompassing the landfill was comprised of approximately 1008 residents and had a higher percentage of Black or African American residents, with 37% in the block group versus 24% in the whole county. Also, the block group has a slightly lower percentage of White residents, with 44% in the block group versus 50% in the county. There was no difference in the percentage of Hispanic residents (20%). Alaska Natives/American Indians represent 1% within the block group of study, and 1.7% in the county, and Asians and Native Hawaiian/Pacific Islander residents represent less than 1% of the population.

Snow Hill is situated approximately nine miles east of Clinton, the county seat, with a low-lying elevation of 144 feet. The landfill is the highest point in the county. The area has alluvial, sandy soil that presents challenges for private well as well as public water infrastructure.

### 2.2. Participants

This photovoice study evolved from a developing partnership between a student researcher from the University of North Carolina at Chapel Hill (UNC-CH) and a community leader actively involved with the North Carolina Environmental Justice Network (NCEJN) who met at an environmental justice (EJ) organizing meeting in a nearby county. The community leader provided input into the project’s design and recruitment process. The photovoice method was selected by these two co-authors in their first meeting. Given the aim of the photovoice method in developing critical consciousness to promote social action [8], the community organizer believed the method would garner support for community mobilization around local EJ issues.

A second student researcher from UNC-CH joined the team to help facilitate the three photovoice discussions. Both student researchers are White and recognized that their backgrounds differ substantially from that of the community partner. Given the history of unfair treatment of African Americans in research, transparency and authentic communication were pivotal in developing a trusting relationship with the photovoice participants and the broader community. The students were supervised and mentored by an African American faculty member from UNC-CH with lived experience in a rural agricultural community and expertise in EJ research and practice. The team made efforts to develop relationships with Snow Hill community members before and after completing the photovoice study. All study procedures were approved by the UNC-CH Institutional Review Board for Human Subjects Research, and all participants provided written informed consent, including parental consent and minor assent for one participant.

Participants were recruited at an NCEJN Quarterly Meeting held in the fall of 2016 in a rural, predominantly African American community in eastern NC and home of the community leader. Attendees were presented with information about the study, its purpose, and what participation in the study would entail. The team used purposive sampling to identify community members interested in EJ activism and convenience sampling to recruit members interested in participating in the photovoice study.

Eligible participants needed to be at least 14 years old and a resident of the Snow Hill community. Participation was voluntary and included a small monetary incentive. Six study participants were recruited, a typical sample size for this small group process [2]. Four participants were present at each photovoice discussion, though it was not always the same group of people. Due to the participatory and exploratory nature of the study, the researchers allowed for flexible participation over the three discussions and opening and closing sessions. At least one participant in the first discussion was present for the second and third. Participants varied by age and professional background, including a high school student, a small business owner, a local minister, and a retired farmer.

### 2.3. Procedure

Photovoice is a systematic research process in which participants brainstorm photo assignments in response to the overarching research topic/questions ((a) How do environmental health concerns in this community impact residents’ perceptions of HRQoL? and (b) how do community and county factors facilitate or inhibit organization around these concerns?), take photos to represent their perspectives on the photo assignments, and collectively review and discuss the photographs using a structured, facilitated process called SHOWED, described below. In this method, photography serves as a universal language for showcasing perspectives often overlooked in traditional research methods [8,47].

The study took place over five sessions. In the first session, the student researchers oriented the participants to the photovoice method and its use in assessing the HRQoL concerns expressed by community members in relation to industries located in their community and in developing actions for addressing raised concerns. The researchers made sure participants understood the process and how the data were to be used, obtained participant consent, and generated the first photo assignment with them. In the second, third, and fourth sessions, the student researchers facilitated the participants in reviewing their photos and discussing the health and quality of life impacts of a large landfill and CAFOs located in their community. For each of these sessions, participants were instructed to bring one photograph that responded to participant-generated photo assignments related to the community’s concerns and voted to use one photo from the group using facilitated discussion.

The discussions were facilitated by the student researchers using the SHOWED dialogue method, which asks participants (1) what do you See in the photograph, (2) what is really Happening, (3) how does it relate to Our lives, (4) Why does this issue exist, (5) how can we become Empowered by our new understanding, and (6) what can we Do about it [7]? The SHOWED discussions were audio-recorded with consent from all community partners involved and analyzed at multiple stages, per the description below.

In the fifth session, the researchers presented action items and community organizing objectives that had emerged from sessions 2–4 back to the participants for a discussion on planning the next steps. The findings from this study and participant-recommended action items were presented to community members and stakeholders at the 2017 North Carolina Environmental Justice Summit.

### 2.4. Data Analysis

The researchers transcribed the audio recordings following each discussion and reviewed data to develop preliminary analytical ideas to be shared with participants at the next discussion. The transcripts were deidentified to remove personal information such as names and identifying characteristics of individuals, groups, and places to maintain participant and community confidentiality. Member checking, a process of taking data and interpretations back to the participants to determine the credibility of the information, was vital in ensuring the correct interpretation of the lived experiences of the participants [48]. After the final session, the researchers conducted an in-depth qualitative analysis of the discussion transcripts and used community feedback on preliminary findings to shape the analysis.

The researchers used Atlas.ti 8 software for coding and analysis [49]. After reading through all three transcripts, session notes, and notes from the 2017 NC EJ Summit, the university researchers pared down a list of potential codes to create a 38-item codebook. The codebook contained 23 topical codes to be applied to data explicitly mentioning topics related to environmental health and hazards and community dynamics, and 15 interpretive codes to be applied to data representing common themes and motifs showing up in discussion. One interpretive code was created using the in vivo coding method and is discussed in the findings. In vivo coding is used to ensure concepts derived from qualitative data stay as close as possible to participants’ own words or to use participant terms that capture a key element of what is being described [50]. The student researchers and faculty supervisor coded the three transcripts and ensured inter-coder reliability by reviewing codes applied to quotations in the first discussion transcript and reconciling any differences in code application between the researchers. The researchers engaged in content analysis by reviewing notes from each of the three photo discussion sessions and code reports produced in Atlas.ti to identify three analytical themes that serve as the key findings presented below.

## 3. Results

A summary of each discussion session, including the date, number of participants, participant-developed title/theme for the session and the discussion photo, is presented in Table 1. The participant-generated photograph and discussion assignments were titled as follows: Session 1: “How are we living?”; Session 2: “Are we living or are we surviving?”; and Session 3: “When does it stop?” At the start of each session, participants shared their photographs and voted on one to be used as the “discussion image” for the facilitated discussion.

One statement from a participant, “it’s a battle”, represented the salient story of the Snow Hill community’s plight and efforts in organizing throughout the discussions and thus was turned into an in vivo interpretive code [50]. The codes used in the content analysis are listed in Table 2, with the number of times each code was applied to the data. Italicized subcode names are listed below the parent code.

The coded data were further organized into three broad themes: (1) health and quality of life concerns, (2) industry influence on community cohesion and self-determination, and (3) actions to address environmental injustice in Sampson County. Respectively, these themes reflect the major environmental hazards and the community’s relationship to them, their impact on the community, and what can be done to address these concerns.

### 3.1. Health and Quality of Life Concerns

In the three discussion sessions, participants shared anecdotes supporting their belief that the residents’ water supply is contaminated with run-off from surrounding CAFOs and the landfill. An image of one participant’s toilet revealed a thin stain forming around the standing water line in the toilet bowl (Table 1). Participants chose this photograph for the first photovoice discussion to highlight the fear of water contamination. The person who captured this photo describes their rationale below:


*Um, well I took [this photo]. I saw something not natural, and if people are seeing a stain on their porcelain, and drinking it, what’s it doing to their insides? It’s gotta be something. I mean, if it’s concentrated enough to cause a stain, then it’s got to be concentrated enough to cause someone to be ill. If not all at once, then over a period of time.*
(Participant 1 (P1))

Participants pointed out that non-residents who see this photograph may claim that the person who owns this toilet was not cleaning it enough and the line was caused by natural algae build-up rather than from industrial byproducts. All agreed that the notion that residents are “crying wolf” about contaminated water supply was a barrier to being taken seriously on water contamination and health concerns.

Most participants described routinely purchasing bottled water for drinking and cooking to avoid consumption of contaminated water, though they realized using tap water and risking exposure was inevitable via daily hygiene activities, washing pets and vehicles, and consuming ice made from tap water. Purchasing bottled water while still paying for their connection to and use of the county water contributed to financial strain for participants and their families.

Participants also expressed concern about exposure to harmful air pollutants emitted from the surrounding industries in their community. They pointed to the frequent, lingering odors that originated from CAFOs and the landfill, which made them fearful of breathing in pollutants. They lamented the fact that they and their families were unable to enjoy being outside on the property they owned, paid taxes on, and claimed as home, a right afforded to most homeowners.

In all three discussions, participants mentioned increasing cancer incidence in the community. They attributed this increasing incidence to water and air pollutants from nearby industries but expressed frustration at their powerlessness to bring about change. Participants believed they would not be taken seriously by some public officials unless they had data connecting industrial hazards to poor health outcomes and worsening quality of life. They expressed a strong need for more knowledge and resources to collect data to establish the connection, saying that such evidence was necessary to demand better management of hazardous materials at nearby industries and environmental protections for the surrounding community. These sentiments are evident in Participant 5′s comments below, occurring at different points in the photovoice discussions:


*And it’s happening in these low-income areas. And that’s not right… I know what we’re talking about with these grants but it seems like it’s going to be a long-term thing. We need some help with this water and this air. I think if we got that going and we had some proof that this is what they’re breathing, this is contaminated, it’s not what it’s supposed to be, we can get things going.*
(Participant 5 (P5))

P5:
*See this is what interests me [holding a pamphlet about environmental justice issues in NC, referencing chemical emissions from CAFO facilities]. That arsenic? It would be good to find out what kind of companies are using this kind of.*


Facilitator 1:
*Yeah, that’s in their food?*


P5:
*Yeah, and then you know what they do with the food. After consumption, they spread it all out on the fields. Those toxins…and they’re talking about spraying on the fields? We don’t think about that as sulfuric acid and all that but that’s what they’re doing.*


Facilitator 1:
*Mum. Right.*


P5:
*All of this is emitted. I think if we use the terms that are…and educate them on the terms. The terms are what make people go, “what? I didn’t know that.” But it’s what they’ve been doing the whole time. They’re spraying this, they’re spraying sulfuric acid, the chemical.*


Participants discussed how nuisances originating from the nearby industries impacted their psychosocial health, including poor mental health and quality of life. As described above, frequent odors from CAFOs and the landfill prevented them from engaging in certain activities, such as hosting outdoor cookouts at home and at church. The inability to connect with family and community in this way has led to feelings of social isolation. In addition, the daily nuisances, fears of becoming ill, and visual reminders of the looming landfill have caused depression symptoms for many residents.

P1:
*There’s a lot of people in Snow Hill that have been treated for depression. Yeah. A lot of us. And it’s really depressing when someone asks you, “how do you get to your house?” and you say, “well turn left at the landfill,” you know? It’s a landmark, you know? I’ve tried to give a little bit better directions, something like, “just go down to this road,” but they get confused and I finally say, “just go down past the landfill.” Yeah. You don’t want people to know you stay by the landfill but that’s the reality.*


F1:
*Is that pretty widespread? Do you know people who have been treated for depression?*


P1:
*Oh yeah.*


P6:
*Quite a few of them.*


The landfill attracts buzzards (also known as turkey vultures) into the community, which are a major disturbance of peace to residents. Participants voiced annoyance and fear over the increasing and consistent presence of buzzards in trees in their yards and on the roofs of houses, 10 of which have been outlined for ease of viewing in the discussion image for the second photovoice discussion below (Figure 1).

During the discussion, participants described the impact of the large buzzards in their community.

P4:
*You don’t feel secure. It messes with your peace, you know. It relates to our lives in such a way—it’s like my situation. You don’t want to bring a whole lot of people around to hang out in your yard. Stuff like that.*


P5:
*It limits your social interaction.*


P7:
*It depletes your quality of life, your right to enjoy your property.*


P5:
*It increases your anxiety, with the buzzards not shooing when you say shoo.*


Crowding of buzzards on the roofs of homes not only instills fear in residents from the loud stomping and scraping noises the birds make, but they also contribute to property destruction as they have destroyed many residents’ roof shingles. This causes an additional financial burden for residents who must replace or repair their roofs. The image in Figure 1 shows buzzards congregating on the roof of the home of a resident who had recently had the roof of their carport restored. Participants shared that shortly after its installation, buzzards destroyed the carport roof by scratching up and eating the shingles. This homeowner and other residents with similar stories face increased costs for home maintenance due to property invasion by buzzards attracted to the community because of the landfill.

### 3.2. Industry Influence on Community Cohesion and Self-Determination

Residents of this community identified the county landfill as the most pressing environmental hazard and threat to the quality of life in their community. Despite suspicion of the landfill’s negative impacts on environmental and human health, our lead community partner had struggled with getting consistent community participation in local EJ efforts. Participants revealed that nonparticipation might be the result of divisive or coercive tactics to prevent residents from speaking out against the industry. These tactics have created a distrustful relationship between some residents in the community and landfill management and county officials.

According to participants, construction of the county landfill began in the early 1970s. The landfill was initially plotted as 40 acres to be filled over the management company’s 20-year lease period. Participants were unclear about who was leasing the land to the landfill company and how additional land continues to be acquired. They speculated that landfill management continues to buy the surrounding land and claim farmland from deceased residents by labeling themselves an agricultural business. Participants stated there are fewer restrictions on agricultural industries in NC than there are on waste management industries.

The third photovoice discussion image shows piles of landfill liners waiting to be used to line new holes for dumping (Table 1). Participants viewed the liners as a symbol of boundless expansion, associating them with the landfill management company’s authority within the community to expand at will. Participants believed that landfill expansion is made possible by manipulation at the cost of community residents’ safety and health to maximize profits. Despite efforts to communicate concerns, participants expressed that unless they have data confirming their concerns to county and state regulators, they lack the power to stop landfill expansion and reveal its detrimental impacts through their experiences alone. When participants had previously spoken out about the issues they were facing, some participants felt unheard, and their concerns were dismissed by the public officials and landfill management. As a result, some community members have become discouraged from engaging further with authority figures to create change. One participant was frustrated by the apathy and lack of compassion they received from one public official in response to their concerns about environmental nuisances resulting from the nearby industry.


*It’s a sad situation, you know? I went and talked to the [public official] …. I don’t know why [they have] the job that [they have]… And when you go try to see [them], [they come] out like, “you know what… I smell it too. I know what you’re going through. I smell the same thing you smell. But until your quarterly meeting and everyone gets engaged and complains about the same thing, my hands are tied.” And I said, “but you’re the [public official]!”*
(Participant 6 (P6))

Participants acknowledged the pattern of historical disenfranchisement of minoritized communities. Many residents of this predominantly African American community are descendants of landholders whose occupancy predates the landfill. Participants felt their rights to the land were being threatened with no authority figure looking out for them or attending to their needs and concerns:


*That landfill doesn’t get that way on its own. It had to be approved somewhere and whoever was supposed to be looking out for [our community], they didn’t do it. They missed the meeting; they just didn’t get the memo.*
(Participant 6 (P6))

Despite their perceptions of the growing impact of environmental hazards on the community, participants expressed a reluctance to leave due to both loyalties to their home/family land and financial constraints. Participants also noted how proximity to the landfill has depreciated property values while simultaneously increasing property taxes, making relocation largely infeasible. Despite their deep ties to this county and community, participants lamented that home is no longer a safe or pleasant place for residents of this community to reside.

Participants also pointed to how decisions on landfill placement and expansion were made without community member representation and consideration. They felt as if their voices had been omitted from these discussions since they would present a counterargument to landfill expansion. In addition to exclusion from development decision-making, participants shared perceptions that some residents may have been encouraged to keep quiet about resentment towards expansion.

Decisions made by landfill management caused tension and division within the community. Participants regarded landfill representatives as selective in whom they communicated with and invited to events and whom they sponsored. Participants also reported that instances such as these have been occurring over the past decade or two as the landfill expansion rate increased exponentially. These divisive tactics present a major barrier to organizing a community group around this issue, as many residents are fearful of retaliation if they openly express their opposition to the landfill.

### 3.3. Actions to Address Environmental Injustice in Sampson County

Photovoice, as an action-oriented research process, is designed to generate community-initiated action items by the participants [8]. Participants in this study identified action items to help address environmental justice concerns in Sampson County, as shown in Table 3.

To address concerns about contaminated water, participants requested that their academic partners aid in connecting them to independent water and air testing analysis to check for unsafe levels of hazardous contaminants from the landfill and CAFOs. They expressed distrust of county reports on water testing, as they may not be conducted frequently enough or might be falsified results that would otherwise implicate these industries contributing to economic development in the region. Just as residents distrust landfill management, they also distrust local government and utilities companies as they may be prioritizing industry over constituent needs.

P2:
*Question is, do they test the water like they are supposed to, or do they just tell you something?*


P1:
*That’s why I’m saying we need separate, independent testing.*


P4:
*Do a comparison.*


Facilitator 2:
*Is that something that you all think you would like to do independent of the County?*


P1:
*Oh yeah, we would love—*


P4:
*Yeah, if somebody else could come over and test it.*


They also recommended increasing community awareness of environmental hazards in Sampson County and their impact on resident health and wellbeing through multiple means, including voter education, distributing facts on health risks of the nearby industry to residents, having face-to-face communication about environmental hazards, and holding presentations for the broader community and local organizations as part of community mobilization to realize environmental justice in Sampson County.

F1:
*Is there a particular format for communicating that information… that would reach people?*


P1:
*Well, meetings if you can get them, but you really need someone to get out and talk to them. You know, [lead community organizer] here sent out flyers and things, and I’m pretty sure some flyers he gave some people are probably still on their desk.*


F1:
*Yeah, I’ve heard that; it seems like face-to-face communication is best.*


P1:
*Yeah, sometimes I get out and ask ‘What you think about it? You got a plan? What are you going to do about it? Are you going to fight with me, or you going to talk about it and stay here and die? Are you going to pull out, or what are you going to do?’ Cause, I need some idea about what people think needs to be done.*


Other action items proposed by participants focused on strategizing to grow community organizing efforts in Sampson County. One suggestion was that a small team of community members volunteer to aid the primary environmental justice organizer in the community to reduce some of the burdens of organizing and building community morale. Participants also suggested that residents need to start showing up to more county commission meetings.

P7:
*So, one of the things that needs to start happening is folks living with these buzzards, they need to start showing up at the county commissioner meetings and bringing it up at the county commissioner meetings every time they open their doors. Just talk about it, you know? You got the space where you can sign up? Whether it’s on the agenda or not, I would bring that up every time. They should not be able to have a meeting without somebody bringing it up.*


P5:
*I think when you have enough people that are affected directly by it.*


P7:
*Yeah. People need to start—it’s not going to change unless we do something about it.*


The participants believe that by having community members repeatedly voice their opinions, commissioners and other local officials will be more likely to listen to and understand these concerns, especially as support behind these issues grows. Participants suggested supporting community residents in running for local office to bring community voices directly to the forefront of decision-making.

## 4. Discussion

A frequent theme arising from the photovoice discussions and in response to the research questions was that of health concerns and general perceptions that the nearby industries present a risk of exposure to industrial contaminants. Studies conducted in collaboration with landfill-impacted communities have revealed similar concerns with air quality and water quality [43,51,52,53]. Further validation comes from studies that describe mental health symptoms such as anxiety, depression, sleep disturbances, elevated stress, and respiratory symptoms among residents in communities impacted by CAFOs or landfills [27,32,33,34,54,55].

Another theme centered on participants’ views of the local industries’ approaches to engaging with the community. Several perceived industry engagements as a barrier to organizing for environmental justice. Residents expressed that various forms of manipulation, coercion, and acts of carelessness by landfill management and public officials contributed to the lack of trust this community has in some systems of power. They also described their perceptions of landfill management’s manipulation of residents and community resources for the purposes of industrial growth and to limit opposition, a practice that has been reported in other industrial communities and among workers [56,57,58]. Other studies have described the use of intimidation tactics to silence community organizing efforts [37,59,60].

Similar to other studies, participants in this study voiced an urgency to take action by raising awareness among other residents. Advocacy to public officials and contributing to public discourse on the permitting and regulation enforcement for polluting industries are essential [61]. The multiple polluting industries located within rural communities of color reflect a pattern of environmental racism all too common in the American South and raises a growing concern for cumulative impacts from exposure to multiple environmental hazards coupled with the absence of salutogenic community features (e.g., proper water and sewerage infrastructure, adequate healthcare access, and adequate transportation access) [37,39,40,62]. Furthermore, the burden of proving adverse impacts from neighboring industries is often placed on impacted residents.

Connection to a broader network of experienced organizers, researchers, and legal counsel to assist with mobilizing residents, gathering data, and interpreting permits and laws is an important asset to communities in advocating for change. Participants appeared motivated, had historical knowledge about the community, and displayed a positive outlook on their future despite ongoing adversity. There is clear camaraderie among the participants and a strong sentiment of care for one another and for other members of the community, including their local faith community, where much of this organizing began. By capitalizing on the existing community cohesion, residents stand a better chance of mobilizing others and collectively developing a plan for action [4,63,64].

There are several study limitations worth noting. First, time was a limiting factor in this study. The research team balanced several time constraints, including the desire to keep each session under 90 min, being limited to conducting the sessions on Saturdays only and aiming to complete all sessions within a three-month period so that participants did not lose momentum and have too much time in between sessions. As a result, it is unclear whether the data reached a point of saturation for most of the themes in just three sessions, though several themes did re-emerge in multiple sessions. Furthermore, more time could have been given to the discussion of participant-recommended action items and planning for the next steps.

Another limitation of the study was that all participants did not attend all three sessions. While at least four participants attended each session, a suitable size for photovoice discussion, having a consistent presence of all participants allows the discussion to build in a way that supports action and increases the likelihood of theme saturation.

Finally, an important limitation that likely influenced the photovoice sessions and the interpretation of the data is the identity and lived experience of research team members and their experience with photovoice. The facilitators were two young White graduate students joined by an older African American community organizer with experience working with the community. While the facilitators had limited experience conducting photovoice, each had experience conducting qualitative research and analyzing qualitative data. Also, despite their limited lived experience in rural, African American, and industry-impacted communities, the presence of the community organizer helped to establish trust between the participants and the research team. Regarding the interpretation of the data, the research lead, who has lived experience in an African American rural agricultural community, worked collaboratively with the student researchers to develop the codebook and assign the codes.

Given the study limitations and lack of generalizability of the data, this study is intended to be evaluated as a pilot project to determine how the photovoice method may work in similarly impacted communities and, thus, could be followed by a larger study that would produce a wider range of data. This study also intended to identify key community concerns for subsequent research investigation, applying a combination of environmental monitoring and qualitative inquiry.

By nature, photovoice is an action-oriented research method and promotes co-learning and critical thinking for developing the capacity to address identified needs [2,65]. The action items recommended by the photovoice participants have provided direction for further research activities supported by the community–academic partnership. At the time of this study, the landfill covered 1351 acres and showed signs of expansion, requiring urgency in community-led efforts to protect the community from increased risk of exposure [66]. In 2018, the researchers and community residents completed a community asset inventory and needs assessment to identify community groups, organizations, and businesses that could support EJ efforts in Sampson County and increase advocacy for environmental regulations. Also, academic partners conducted surface water quality testing near the landfill to assess the extent to which contaminants were migrating into the surrounding environment [67]. In 2019, a grassroots organization called Environmental Justice Community Action Network (EJCAN) was developed by local leaders with the goal of increasing community organizing and education around EJ issues within the county. In addition to hosting monthly meetings of residents, EJCAN has collaborated successfully on water and air quality research in Sampson County. They have contributed to public comments and legal actions for stronger regulations on the hog industry, and they have led discussions with local, state and federal policymakers to raise greater awareness and drive action on environmental issues in rural communities.

## 5. Conclusions

This study demonstrates the utility and appropriateness of photovoice as a tool to illuminate ongoing and emerging environmental health concerns in a rural EJ community. Priority areas of focus and future action to contribute toward community organizing for environmental justice were set by the community members represented in this study. Since the completion of the study, the university researchers have continued to partner with the Snow Hill community and residents of Sampson County to conduct water testing independently of the county and nearby industries. Also, local residents established a community-based organization, Environmental Justice Community Action Network (EJCAN), to facilitate community organizing efforts in the county. Photovoice can serve researchers by helping them assess the research needs and interests of a community and can serve community organizers by providing residents with a structured way to discuss their lived experiences and strategize ways to mitigate or eliminate hazard exposure. This study underscores the need for more community-driven research into the cumulative effects of environmental hazards in rural communities. The findings also highlight the need for community engagement by public officials with their rural constituents to build trust and collaborate on the action.

## Figures and Tables

**Figure 1 ijerph-20-05656-f001:**
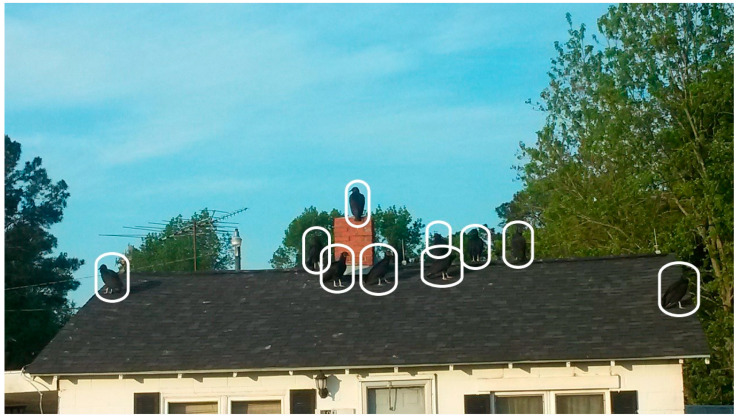
Discussion Image 2: Buzzards line Sampson County Resident’s Home.

**Table 1 ijerph-20-05656-t001:** Photovoice Discussion Session Participants, Theme and Photograph Selection.

Photovoice Discussion Session	Date	Number of Community Members in Attendance	Theme for Photograph Selection and Discussion	Discussion Image
1	27 May 2017	4	How are we living?	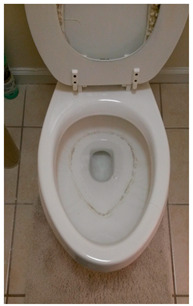
2	17 June 2017	3 + 1 NCEJN co-director	Are we living or are we surviving?	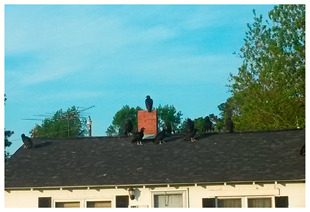
3	15 July 2017	4	When does it stop?	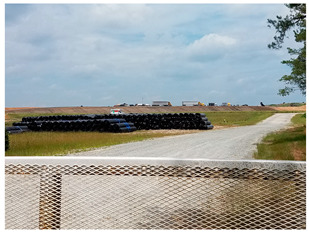

**Table 2 ijerph-20-05656-t002:** List of researcher-generated codes for qualitative analysis.

Code and Subcode Names	Code Type	Code and Subcode Frequency
Take Action	Topical	32
*Need the facts*		10
*Ally with similar/nearby communities*		7
Buzzards	Topical	28
Landfill expansion	Topical	25
Health concerns	Topical	24
*For future generations*		7
*Mental health/depression*		4
Property	Topical	24
*Property invasion/buyout*		13
*Property damage*		6
*Property depreciation*		2
Water Quality	Topical	19
Home	Interpretive	16
*Dissatisfaction with home*		9
Fear	Interpretive	14
Unheard	Interpretive	14
Motivation	Interpretive	13
Quality of life	Interpretive	12
“It’s a battle”	Interpretive/In vivo	11
CAFO	Topical	10
County Services/Officials	Topical	10
*Lack of trust in public officials*		6
Proximity to landfill	Topical	10
Waste	Topical	10
Community	Interpretive	9
*Community resiliency*		3
Resentment	Interpretive	9
Financial burden	Topical	7
Odors	Topical	7
Water testing	Topical	7
Manipulation of systems	Interpretive	6
Coercion	Interpretive	5
Lack of participation from the community	Topical	5
Nostalgia	Interpretive	4
Faith	Interpretive	3
Justice	Interpretive	3
Trucks	Topical	1

**Table 3 ijerph-20-05656-t003:** Participant Recommendations to Address Environmental Injustices.

Independent water testing Air testing Factsheets for residents on the health and wellbeing risks of nearby industries Voter education Face-to-face communication with residents on environmental hazards Presentations; partner with other communities and organizations Media advocacy Involvement in local politics Attendance at county commissioners’ meetings Sharing the responsibilities of organizing tasks/efforts with a small team

## Data Availability

The anonymized data presented in this study are available on request from the corresponding author. The data are not publicly available due to ethical considerations for the anonymity of the study participants.

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
