# Peer review of "Photovoice Reveals Residents’ Concerns for Air and Water Quality in Industry-Impacted Rural Community"

_ijerph, 2023, doi:10.3390/ijerph20095656_

Round 1

Reviewer 1 Report

The detailed comments are in the original manuscript. Further, the novelty of this study was not presented. Conclusions must be rewritten.

Reviewer 2 Report

The article „Photovoice Reveals Residents' Concerns for Air and Water Quality in Industry-Impacted Community“ is very interesting and innovative because of its methodological approach.

The aim of the article is – to use the photovoice method and to assess the health-related quality of life impact of confined animal feeding operations and the Sampson County Regional Landfill on residents of a predominantly African American community in Sampson County, NC.

Two research questions are developed: (a) How do environmental health concerns in this community impact residents’ perceptions of health-related quality of life? and, (b) how do community and county factors facilitate or inhibit organization around these concerns?

The article is very well structured.  It describes the used methodology, presents obtained data, and after, the discussion part and conclusions follow.

The most interesting part of the article is the photovoice method. It allowed the community members to reflect on images and provide unique insights into community challenges and strengths.

The conducted research also has certain shortcomings, primarily due to its scope in terms of respondents and time, but the researchers understand these shortcomings perfectly, they clarified and described them. Therefore, the obtained results can hardly be considered reliable.

However, the study itself can be evaluated as a pilot in order to determine how it works in the community. It could be followed by a bigger study that would provide a wider range of data.

The article is very interesting, written in clear and easy-to-read language.

Reviewer 3 Report

The paper aims to outline an important issue concerning the health-related quality of life. The authors use a qualitative research method, namely photovoice method.

I would like to make the following remarks:

- line 108: the acronym UNC should be firstly explained

- the authors specify that this study concerns  a predominantly African American community in rural North Carolina. However, in Section 2.1. it is underlined that 37 percent of the population under study is Black or African American. The authors should clarify this issue. The structure of the population by area of residence may add some important information.

- In the Conclusions section, the authors should underline the impact of this research on short and long term on the community under study.

Round 2

Reviewer 1 Report

Novelty is missing. Conclusion part needs serious revision.

Round 3

Reviewer 1 Report

The authors made sufficient changes. I recommend publishing the manuscript in current state.